# Does Participation in the "Grain for Green Program" Change the Status of Rural Men and Women? An Empirical Study of Northeast China

Yifei Zhu * and Keshav Lall Maharjan

Graduate School of Humanities and Social Sciences, Hiroshima University, 1-1-1 Kagamiyama, Higashi-Hiroshima 739-8524, Japan; mkeshav@hiroshima-u.ac.jp
* Correspondence: xiu17hi23@gmail.com

**Abstract:** The Grain for Green (GfG) program is an afforestation project created by the Chinese Government to protect the environment. Farmers who participate in GfG return farmland to forest. Losing arable land means losing an income source, so farmers have to reorient their livelihood strategies, leading to potential changes in the gender division of labor. To assess gender differences in the impact of policies, we use indicators from the Women's Empowerment in Agriculture Index (WEAI), which measures the status of women relative to that of men. Using sex-disaggregated data from farmers in mountainous areas of northeast China and applying the inverse probability weighted regression adjustment (IPWRA) estimator, we found that the status of men and women had increased with GfG participation, but women's status had improved more than men's. However, this was not because of their smooth participation in the program, but because of its challenges. Their decision-making skills improved unexpectedly due to pressure to protect their interests. Rural women worried about their families' livelihoods, so they tried to improve their family welfare and diversified their income sources. In this process, women had more interactions with outside communities. Our results underline the strong need to continuously monitor the gender impacts of environmental policies.

**Keywords:** grain for green; women's empowerment in agriculture index; women's status; northeast China; inverse probability weighted regression adjustment; sex-disaggregated data

## 1. Introduction

Historically, it is rare to find the gender–environment nexus in national policy debates on gender equality and environmental sustainability [1]. There are two possible reasons for this. The first is because climate change is often seen as a technical and scientific problem, which has hampered the integration of gender equality and human rights into eco-conservation policy [2]. Furthermore, it is usually examined through the lens of the male-dominated sectors of scientific research and government policy [3]. However, climate change has significant social effects [2,4,5], suggesting that it is necessary to consider gender issues when designing climate-related policies. Failure to do this may threaten the sustainability of the project. For example, due to the gender divisions of labor, rural women have a lot of interactions with nature to meet their subsistence needs such as water and fuel [6,7]. Changes in the climate or environmental policy can affect women's livelihoods. As Ref. [8] found, the failure of development interventions on forest management is due not only to deficiencies in technical and scientific knowledge, but also to social causes. The second reason is that, although sex-disaggregated data collection is being advocated internationally, such data is very limited and continues to pose a challenge for policy makers [1,9–11].

However, this picture has changed in recent years, with growing calls for a more gendered perspective to be integrated into environmental policies at both global and national levels. For example, the Commission on the Status of Women (CSW), which

works to accelerate the realization of gender equality and the empowerment of women [12], selected the following priority theme for 2022:

*Achieving gender equality and the empowerment of all women and girls in the context of climate change, environmental and disaster risk reduction policies and programmes.*

The empowerment of women has been recognized as the central issue in determining the status of women [13], and also as being intrinsically linked to achieving sustainable development [14]. Research has shown that women's empowerment has a positive relationship with stronger environmental policies and environmental sustainability [15,16]. Women's empowerment involves control over non-material resources and ideology [17], not just agricultural production and subsistence needs. Ref. [15] found that women have an increased social and political agency, and that their interactions with other factors can modify the gender relations in society. As a result, women's political participation can effectively adjust the implementation of environmental policies. In Africa, Ref. [16] observed that women's socioeconomic empowerment is not isolated from Foreign Direct Investment (FDI) and GDP per capita, which may influence environmental sustainability. Furthermore, in many societies, women are responsible for raising the next generation, so women's thoughts and behaviors play an important role in the rural ecological environment [18]. The results of some studies concur with the theory of [17] that knowledge, attitudes, and behaviors are some aspects which affect the outcomes of women's empowerment in environmental policy. These aspects help us to understand the correlative mechanisms regarding women's empowerment and environmental policies/programs.

So far, we have only mentioned studies on women and the environment. Nevertheless, the use of the single concept of women's empowerment is not enough to understand the relationship with environmental policies. Ref. [7] indicated that women's environmental relations cannot be understood in isolation from men's because of gender differentials in historical, spatial, and temporal dimensions. Studies have shown that strategies to address gender gaps in sustainable ecosystem management should match the context of the each development program [8,19]. These show that, to understand gender gaps, women's status should be studied in both absolute terms and in terms relative to men [20] according to local conditions. Ref. [21] measured women's statuses in China in terms of income, occupation, and education compared with men; Ref. [22] studied the status of women and girls in Guinea on barriers across all dimensions of well-being relative to men and boys; and the reports such as Status of Women in the United Nations System [23] and Current Status and Challenges of Gender Equality in Japan [24] combined men's data with women's to monitor equal representation of women at all levels of the domestic system.

*Study Context*

The Grain for Green (GfG) Program, also known as China's Conversion of Cropland to Forests Program (CCFP) or the "Sloping Land Conversion Program", is the largest Payment for Environmental Services (PES) program in the world [25–27]. Its primary goal is to reduce natural disasters by mobilizing farmers to convert their croplands on steep slopes and otherwise ecologically sensitive areas into forests [28]. In the 13th Five-year Plan for Economic and Social Development of the People's Republic of China (2016–2020) [29], farmland that meets the standards for GfG is defined as follows:

*Turn farmland that occupies slopes of 25 degrees or steeper affected by serious desertification, or occupying slopes of 15 to 25 degrees in areas of key water sources, into forest or grassland.*

Farmers who meet this criterion can participate in the project on a voluntary basis. Once farmers choose to participate, they give up their part or all of their farmland and therefore lose farm income. The state gives ecological compensation (eco-compensation) contract to farmers according to the loss of farmland. However, the eco-compensation itself hardly covers farmers' opportunity costs [30], leading them to diversify their livelihood strategies by finding other sources of income to make up for lost farm income. The GfG

policy has therefore brought significant changes in farmers' livelihood patterns. Within the context of ecologically fragile regions and mountainous areas, finding effective long-term policies is of great concern. In particular, it is important to investigate how such policies affect people living in these areas, especially rural women. Several major program-related national level environmental policies such as Regulations on Converting Farmland to Forests [31] and Development Plan for Forest and Grass Industry (2021–2025) [32] do not explicitly reference the gender–environment nexus or include policies to protect women's rights.

Due to the deficiency of gender perspectives in the formulation and implementation of the GfG Program in China, women's social and economic conditions have been neglected. For example, even if women get eco-compensation from the program and participate in new survival strategies, it is still difficult for them to transform their livelihoods [30]. Ref. [33] conducted a tree-planting field trial that gave eco-compensation to both men and women, and found that men benefitted more than women, although women provided more effort than men in eco-compensation contract implementation. Plausible explanations of different environmental performances between men and women can be intra-household dynamics such as decision-making, roles, and inequality in labor provided [33].

At the same time, there are different views on women's rights and performances in the GfG program in the literature. A large proportion of the rural male labor force migrated to cities for work, partly as a result of the gender stereotype that "men work outside the home, and women stay at home". GfG further promoted this phenomenon. Due to the differentiated division of labor, almost all tree planting and management have been done by women. On the negative side, it put more pressure on rural women's lives and labor [18,34]. Women worked hard to guard their husbands' forestland, but their rights for forest management were not guaranteed [34,35]. On the one hand, women do not have independent land contract rights [34], and women's labor is not valued by people [35]. Looking on the bright side, after participation of the GfG, women's decision-making power was found to get stronger, because natural resources controlled by women increased and they were more involved in the public affairs of the community [35].

However, most studies mentioned above use qualitative data collected mainly from women. Quantitative data focusing on women and men is needed. More specifically, there do not seem to be any studies which use sex-disaggregated data to explore the causality between China's ecosystem policy and the status of rural men and women. Causal links could point out that women's empowerment is an important aspect in the study of the socio-economic impacts of environmental policies [15].

In this regard, this study attempts to use correlation and causality to examine how the GfG affected the status of rural women and men. It tries to answer two main research questions: first, how has the GfG influenced the status of rural men and women on average? Second, how has the GfG influenced the status of rural men and women when looked at separately?

The remainder of the paper is structured as follows. Section 2 describes the models and variables tested in this study and the methodology used in this study. In Section 3, we test whether participation in the GfG Program had a causal effect on the status of rural men and women, analyzing each group separately. The reliability of the causal inference between GfG and the status of men and women are checked using propensity score matching with calipers. Meanwhile, farmers' statements in interviews are used to explain our results and help answer the two research questions. In Section 4, we describe the implications of the GfG program and connect the empirical results with some of the theoretical motivations of environmental policies. Other factors associated with the change of rural men and women are also mentioned. Section 5 then proposes recommendations on how to better integrate gender into current policies on the environment and ecosystem, as well as on future directions of research.

## 2. Materials and Methods

### 2.1. Study Sites

In order to improve the quality of the environment, China has implemented ecological conservation projects in different areas. Keeping in mind the Outline of the 14th Five-Year Plan (2021–2025) for National Economic and Social Development and Vision 2035 of the People's Republic of China [36], we selected northeast China as the study site for this research. The entire document mentioned northeast China eight times. The three northeast provinces (Jilin, Liaoning, Heilongjiang) are all major grain producers in China, benefiting from their unique geographical advantage. However, the large increase of grain producing areas has resulted in a substantial decline in forest land, which is the main reason for the ecological deterioration of the region [37]. Given the ecological capacity of northeast China, it is important to reverse the malignant trends in the regional ecosystem and to strengthen cooperation in environmental and ecological projects [38]. Consequently, the revitalization efforts for northeast China should prioritize the GfG Program, not solely to optimize the environment but also to retain a harmonious society.

Figure 1 shows the layout of major protection and restoration projects in China. The thick green dotted line on the upper right is the Northeast Forest Belt, formed by the Xing'an Mountains and the Changbai Mountains. The Changbai Mountains are also known as the "dragon vein" in China. Following the Changbai Mountain Range, we conducted our survey in 16 localities deep in the mountains of three provinces: Liaoning, Jilin and Heilongjiang. Figure 2 shows the three provinces within the vector map of China (indicated by red stars), while the study sites are shown as red dots on the satellite maps of the three provinces (shown here at different scales for ease of reference).

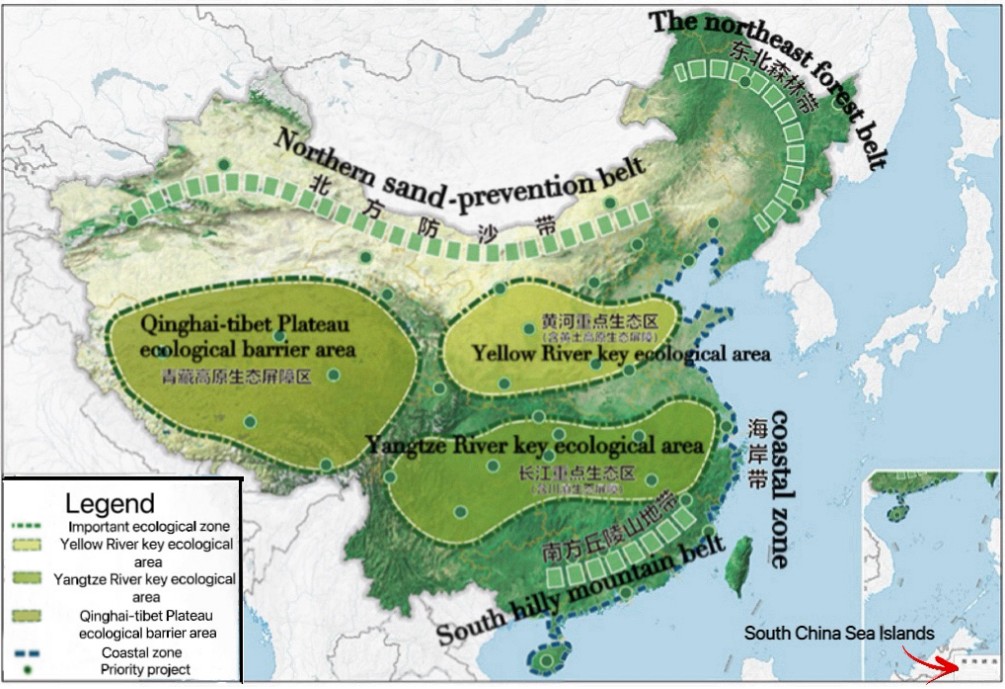

**Figure 1.** Layout of major ecosystem protection and restoration projects. Source: [36].

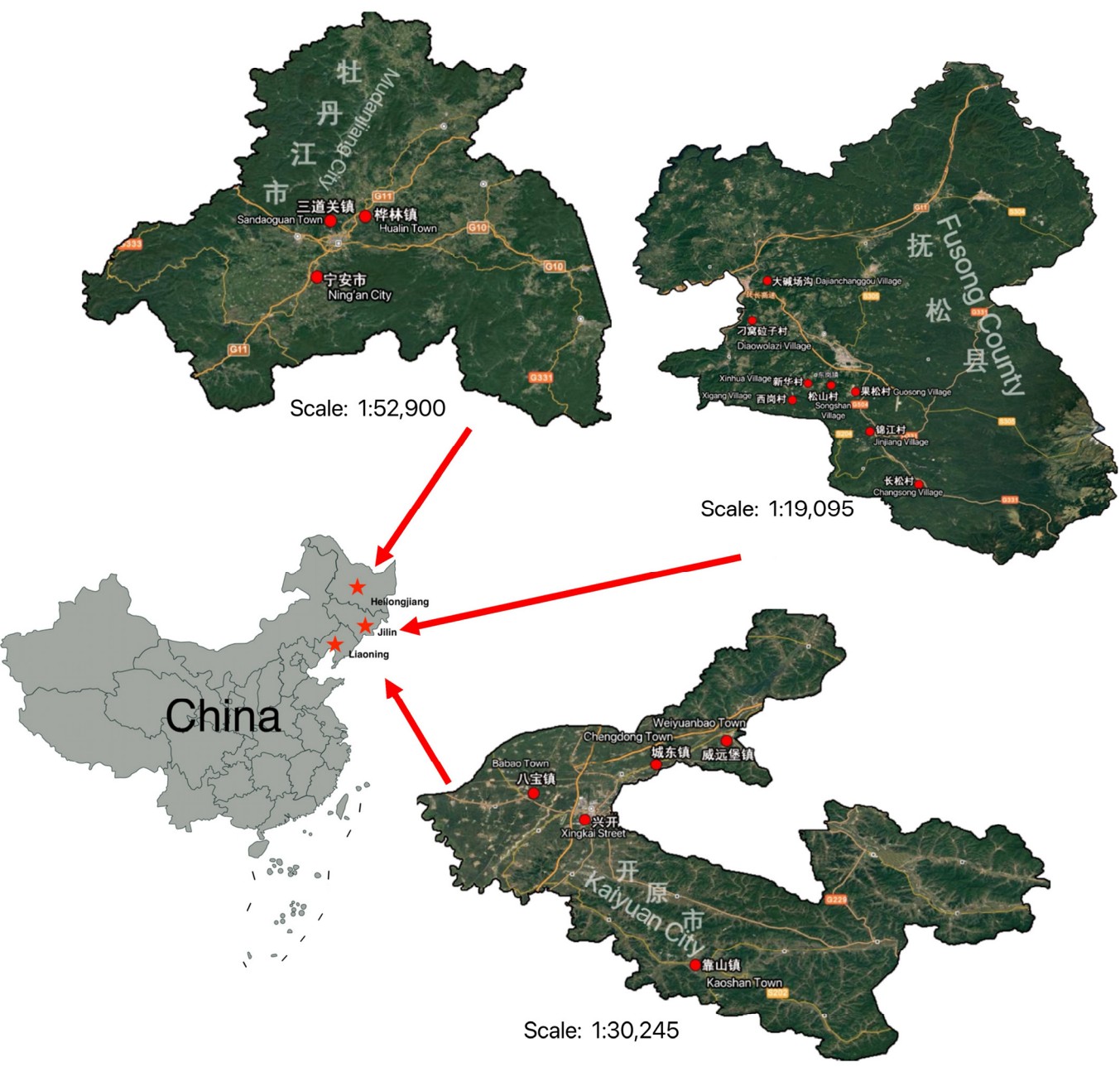

**Figure 2.** Study sites. Source: Authors.

*2.2. Model Description and Data*

2.2.1. Sampling Procedure

Data was collected from April to July 2021 using a multistage sampling technique involving purposive sampling of regions and villages and random sampling of households. Regions within northeast China, and then villages within regions, were purposely selected based on their topography (namely, whether they were suitable for the program to return farmland to forest). Finally, rural households were randomly drawn from the selected villages. It is worth noting that for the vast territory of Heilongjiang Province, the sample was too small to select only one county/county-level city/municipal district (county, county-level city and municipal district are treated as having the same administrative status in China). For example, "Hualin Town" (seen at top left of Figure 2) is a town under the jurisdiction of Yangming District, Mudanjiang City in Heilongjiang Province. However, Yangming District covers only 3.45% of Mudanjiang City's total area and is so small that it would have had to be combined with adjacent geographical units in order to

qualify satisfactorily as a primary sampling unit [39]. Thus, three localities were selected in Heilongjiang Province (Ning'an County-level City, Yangming Municipal District, and Aimin Municipal District under Mudanjiang City); eight localities in Fusong County under Baishan City in Jilin Province; and five localities in Kaiyuan County-level City under Tieling City in Liaoning Province. In Figure 2, we show one prefecture-level city (Mudanjiang City), one county (Fusong County) and one county-level city (Kaiyuan City) (for ease of location).

In the literature, the household head usually refers to the individual who earns most of the family income [40], and who is acknowledged as the head of the unit by other members or by himself/herself, if living alone [41]. However, in recent studies, there may be more than one household head. In [42], the author systematically examined household heads among married women, and demonstrated that married women could be deemed as "head of household". From this, the heads of households in rural China can be roughly divided into three cases: registered couples under the same roof (male household head and female household head); unmarried men or women living alone (male household head or female household head); or divorced men or women with their children or parents under the same roof (male household head or/and female household head).

Keeping the above in mind, we drew a cross-sectional sample of 200 household heads of a total of 112 rural households, or approximately 0.097% of the population of the 16 localities. Due to the costs, resources, and time constraints, conducting surveys in the mountains and valleys across the vast northeastern provinces is difficult. Thus, we used "design effect" to obtain an effective sample size. To get an effective sample size we collected data on parameters such as total population size, cost per unit, sample size under optimal allocation, sample mean, and so on [39]. Based on this research context, we got 59 samples in Liaoning province, 83 samples in Jilin province, and 58 samples in Heilongjiang province. In the example offered by [39], a population size of 1,000,000 needs 1705 effective samples, or around 0.17% of the total population.

Out of the 200 samples covering the three types of household heads, 191 people are registered under the rural hukou ("农村户口/nóng cūn hù kǒu/" in Chinese), while the remaining 9 people have non-rural household registrations, also known as urban hukou ("城镇户口/chéng zhèn hù kǒu/" in Chinese). The sample also captured a variety of GfG conditions (e.g., state-owned forest, collectively-owned forest) and information about household welfare, including the labor participation of each household head. Although the sample size was small relative to the total population, it can be said to represent regional rural life as our samples covered vast areas in the sparsely populated rough terrain, capturing minute differences within the areas.

The survey had 60 questions divided into four parts: (1) demographic information; (2) five domains of empowerment (5DE) in agriculture including production, resources, income, leadership, and time; (3) state of diversification; and (4) fundamental state of GfG Program. Questions were designed to provide a better understanding on empowerment, agency, and the inclusion of men/women in the agricultural sector under the GfG Program. Based on respondents' answers to the paper-based questionnaires, a few follow-up questions were added to get additional information. Each respondent was interviewed alone, with the interviews lasting 60 min on average. All 200 respondents agreed to both structured and semi-structured face-to-face interviews. Data were gathered from the field, respondents' homes, or village meeting rooms. Before the survey, the authors got the respondents' consent and taught them how to fill out the questionnaires (for the respondents who were not willing to write the questionnaires but accepted our survey, the questionnaires were filled out by the authors). If there were questions that respondents did not understand, authors would explain them one by one. This kind of omission was filled out by authors (manual editing). Thus, response rates were 100%. The authors checked the quality during the survey. After the survey, any uncertainties or doubts to answers were confirmed with respondents by electronic communication such as WeChat.

2.2.2. Estimation of Women's Status

We assessed rural women's statuses by adopting a method for calculating the inadequacy score in the Women's Empowerment in Agriculture Index (WEAI) [43]. There are limited tools available to measure the impact of agricultural interventions on women's empowerment [43]. Ref. [15] used Women's Political Empowerment Index (WPEl) to reflect the level of women's participation in decision-making in society. Ref. [16] adopted the "Women, Business and the Law Index" to examine women's socioeconomic empowerment in the context of women's prospects as entrepreneurs and employees. However, these indices are not applicable to rural household surveys. In other words, the WEAI is most suitable for analyzing micro-data reflecting the agency of men and women in the agriculture sector. As the inadequacy score in the WEAI can also be applied to men, we were able to compare women's status with men's status.

There are two steps to calculate the inadequacy score. Based on answers to the survey questions in the five WEAI domains (5DE, shown in the first column in Table 1), we first analysed the set of answers for each indicator to examine whether respondents had achieved the 5DE (second column in Table 1). The results for each indicator were dichotomous variables (0 = achieved or 1 = did not achieve) based on a given cut-off level, while the second step involved the weighting and calculation of the respondents' inadequacy scores (third column in Table 1). The inadequacy score is a fractional outcome which lies between 0 (not inadequate) to 1 (fully inadequate):

$$s_i = w_1 I_{1i} + w_2 I_{2i} + \cdots + w_d I_{di}, \quad s_i \in [0, 1] \tag{1}$$

$$\sum_{d=1}^{10} w_d = 1 \tag{2}$$

where $s_i$ is inadequacy score of individual i, and $w_d$ is the weight multiplied by the ten indicators $I_{di}$ that are dichotomous variables. The sum of the weights is 1. The higher the score, the lower the status.

**Table 1.** Domains, indicators, and weights in the Women's Empowerment in Agriculture Index. Source: [43].

| Domain | Indicator ($I_{di}$) | Weight ($w_d$) |
|---|---|---|
| Production | Input in productive decisions | 1/10 |
| | Autonomy in production | 1/10 |
| Resources | Ownership of assets | 1/15 |
| | Purchase, sale, or transfer of assets | 1/15 |
| | Access to and decisions about credit | 1/15 |
| Income | Control over use of income | 1/5 |
| Leadership | Group member | 1/10 |
| | Speaking in public | 1/10 |
| Time | Workload | 1/10 |
| | Leisure | 1/10 |

Some aspects of our methodology differed from [43]. First, while the WEAI does not directly mention the household head when choosing samples, for our study we collected data from household heads because the children's parents are usually at home alone in rural China. The advantage of this approach is that the data comes from the representatives of the household units and is therefore likely to be more reliable. Second, in the WEAI, respondents who are not involved in agriculture might be judged as disempowered [43]. In our case, we treated employed workers with a regular salary or who were never involved in the management of farming as missing values. There were 23 cases with missing values for inadequacy score (1.15% of the sample). Thus, our sample size was 177. Third, as our aim was not to examine the individual indicators in Table 1, we do not go into detailed explanations of each indicator's results in this study.

We also collected data for several indicators which could be very subjective, as Table 1 shows. For example, "input in productive decisions" was asked to all respondents, even farmers who had stopped farming and transferred their land to others. These farmers were allowed to respond based on their previous experiences. Another was "autonomy in production", measured by the relative autonomy indicator (RAI), which is a precise number originally designed by [44] for expressing women's action under coercive or internalized social pressure (though we did not investigate this point in depth as it was not our focus). Thirdly, we looked at "Leisure" by asking respondents to rate their satisfaction for leisure activities. Respondents were asked to rate their satisfaction from 1 (not satisfied) to 10 (very satisfied) using a smiley face rating system [45] (Figure 3).

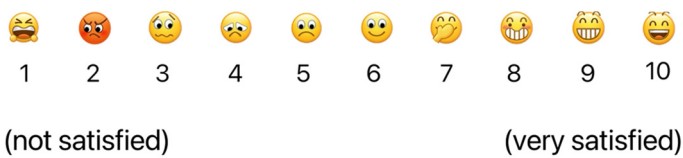

**Figure 3.** Smiley face rating system. Source: [45].

2.2.3. Treatment Group and Control Variables

The treatment GfG was a dichotomous variable of 0 for respondents who did not join GfG Program, and 1 for treatment group who joined GfG program. It had 9 missing variables, as 9 non-rural hukou people did not have land to contract out to the GfG Program.

For control variables, we chose gender, marital status, number of children, education, age, family members (excluding the respondent), diversification, diversification income, house in town, and contracted land. We used both the number of children and family members (excluding the respondent) as the control variables. The two variables have no multicollinearity as the numbers of children are not included in family members. Most children born in rural China immigrate to the city to study or work, and few stay in their hometowns. The number of children affects both men's and women's consumption decisions, such as buying a house in town to give their children a better education. The ownership of the house may also influence their access to modern facilities, which in turn affects other behaviors embodied in inadequacy scores. Ref. [46] found that many rural families rent or buy houses to accompany their children to study in the city from kindergarten and primary school, so that their children can enjoy the high-quality education resources of the city.

Contracted land means farmers transfer their land use rights to others. Farmers transferred the land for off-farm activities. Shifting from farming income to wage income can raise the income of rural households [47,48]. When men and women experienced a change of income and a change in the source of income, it affected their decisions on things like purchasing decisions and time. To some extent, it will affect their status across the 10 indicators given in Table 1. When farmers' lifestyles change, diversification is a necessary control factor. Diversification in rural China means an alternative income source that may be self styled and time flexible, including rural side jobs, part-time jobs, entrepreneurship, and so on. Ref. [49] describes the process by which rural families construct a diverse portfolio of activities in order to survive and improve their standards of living. Diversification is an expected livelihood strategy for both GfG participants and non-participants, which may change farmers' status. Here, we included diversification income as well. We use units of China Yuan (1 Yuan ≈ 0.14 US Dollars) for the variable on monetary changes, such as diversification income. All variables are listed in Table 2.

**Table 2.** List of all variables with descriptions.

| Variable Name | Variable Type | Description |
|---|---|---|
| Outcome variable | | |
| Inadequacy score | Continuous variable | Estimation of status of rural men and women |
| Treatment variable | | |
| GfG | Dichotomous variable | Participation in GfG program (0 = do not participate, 1 = participate) |
| Control variables | | |
| Gender | Dichotomous variable | Sex of the respondent (0 = female, 1 = male) |
| Marriage | Dichotomous variable | Marital status of respondent (0 = others including widowed, divorced or single, 1 = currently married) |
| Number of children | Continuous variable | Number of children |
| Years of education | Continuous variable | Years of education in school |
| Age | Continuous variable | Age of the respondent |
| Family members excluding respondent | Continuous variable | Current family size Number of people living with respondent now |
| Diversification | Dichotomous variable | Whether respondent has a diversified livelihood (0 = no, 1 = yes) |
| Diversification income | Continuous variable | Earnings from diversified sources |
| House in town | Dichotomous variable | Owns house property in town (0 = no, 1 = yes) |
| Contracted land | Dichotomous variable | Whether respondent contracted out their land to other farmers or not (0 = no, 1 = yes) |

### 2.2.4. Data Analytical Method

Fractional Response Model (FRM)

The statistical analysis was done using Stata 16. We first estimated fractional response models for 177 respondents to check how the treatment and control variables correlate with the outcome. The inadequacy scores of the respondents fall primarily between zero and one, giving consistent estimates of the parameters of the conditional mean. The purpose is to check whether the independent variable GfG is associated with the dependent variable, the inadequacy score.

$$E(y_i|x_i) = G(x_i\beta) \tag{3}$$

where $y_i = s_i$, $0 \leq y_i \leq 1$, $Z \in \mathbb{R}$ and $G(\cdot)$ is a standard normal cumulative distribution function (cdf). We assume that Equation (3) is a linear model for the inadequacy score and depends on a $1 \times K$ vector of explanatory variables including GfG, gender, marital status, number of children, education, age, family members (excluding the respondent), diversification, diversification income, house in town, and contracted land. After the model was regressed, we used margins to obtain elasticities, as the functional form at the core of the models is not defined for independent variables that are equal to 0 and/or 1 [50].

Inverse Probability Weighted Regression Adjustment (IPWRA)

Potential outcomes were estimated using the IPWRA approach developed by Wooldridge [51,52]. IPWRA estimators are doubly robust, combining two approaches to estimate the causal effect of an exposure on an outcome [53]. Therefore, regardless of whether the treatment model or the outcome model is mistakenly specified, the effects can still be consistently estimated. We used IPWRA estimators to account for the non-random treatment assignment and examine the impact of policies or programs such as GfG, to make the results more convincing. The rationale for adopting IPWRA is based on methodological and practical reasons.

First, the methodology features the combination of inverse probability weighting (IPW) and regression adjustment (RA). More specifically, the three steps were as follows. First, for the treatment model, a logit regression including the control variables was used to predict the probability of an observation being treated. Subsequently, the sample was reweighted

by inversing the predicted possibilities to get the true ATE. Ref. [54] used IPWRA to explore the effects of social service contacts on teenagers in England because this helped them to identify the true value of a particular intervention and a counterfactual estimation. Next, for the (weighted) outcome model (and in the same manner as FRM as shown in Equation (3)), the potential outcome is estimated for each observation. Equation (3) had all the control variables run twice for the sample being treated and not being treated under the condition of IPW, and the RA is performed in this step. Then, the difference between the calculated average outcomes for GfG (treated) group and the non-GfG (untreated) group is the estimated treatment effect.

The second reason for adopting IPWRA is that the de facto application of our collected data to other powerful causal models like instrumental variables (IV) and regression discontinuity designs (RDD) is not appropriate. Ref. [55] faced the same applicability problem, so they employed IPWRA with average treatment effects on the treated (ATT). Their method estimated the expected average effects of adopting a conservation agriculture (CA) technology option compared with the alternative of non-adoption of a CA technology. ATT is not suitable in our case, because our respondents are almost divided equally between the treated and untreated groups.

Robustness Check

For the sake of finding the true ATE, IPWRA expands non-existent observations serving as a stopgap, which may cause some biases. In addition, for our second research question, we ran men and women's results separately, resulting in bigger standard errors and lower degrees of freedom. To solve the limitations in IPWRA, we first used caliper to find common support (CS). Second, we looked at the coefficient of the interaction term:

$$
\begin{aligned}
s_i = {} & \beta_0 + \delta_0(\text{gfg}) + \delta_1(\text{gender})(\text{gfg}) + \delta_2(\text{gender}) + \beta_0(\text{age}) + \beta_1(\text{education}) + \beta_2(\text{diversification}) \\
& + \beta_3(\text{diversification income}) + \beta_4(\text{marriage}) + \beta_5(\text{family members except respondents}) \\
& + \beta_6(\text{children number}) + \beta_7(\text{house in town}) + \beta_8(\text{contracted land}) + u
\end{aligned} \tag{4}
$$

where $s_i \in [0, 1]$ and $\delta_1$ shows the difference of gender-specific average treatment effects (ATE). This modelling strategy provides the causal effect on the population who conducted GfG. It can be defined as $ATE = E(Y^1 - Y^0)$, where $Y^1$ is the outcome of the treatment group, and $Y^0$ is the outcome of the untreated. These results are given below where we discuss the robustness check. Ref. [56] used density functions to check CS before using IPWRA, which also provided the robustness.

To get more precise estimates of treatment effects, baseline data on outcomes can be matched [56]. Our approach, however, was limited by the absence of baseline data such as the status of rural men and women before the survey, or other pre-treatment variables, such as respondents' ages, education years, and so on, before the GfG. Regardless of how strong the causality shown was, our results were based entirely on the conditional independence assumption (CIA). Ref. [56] also justified this feature with IPWRA.

## 3. Results

### 3.1. Sociodemographic Characteristics and Inadequacy Scores of Farmers

3.1.1. Descriptive Statistics about the Collected Data

The mean and standard deviation of all variables collected in the three provinces are presented in Table 3. Although we collected a total of two hundred samples, nine were non-rural hukou. Non-rural hukou means that no farmland has been allocated by the state, thus no farmland can be returned to forest. Excluding these nine, one hundred and ninety-one people have the right to decide whether to participate in the GfG or not. From this situation, 87 respondents chose to participate in the GfG, while 104 respondents chose not to. In other words, more than half did not participate in the program.

**Table 3.** Descriptive statistics of the collected variables.

|  | Min. | Max. | Mean | Std Dev. | Obs. |
|---|---|---|---|---|---|
| GfG | 0 | 1 | 0.455 | 0.499 | 191 |
| Contracted land | 0 | 1 | 0.408 | 0.493 | 191 |
| House in town | 0 | 1 | 0.355 | 0.480 | 200 |
| Gender | 0 | 1 | 0.520 | 0.501 | 200 |
| Marriage | 0 | 1 | 0.915 | 0.280 | 200 |
| Number of children | 0 | 4 | 1.705 | 0.896 | 200 |
| Age | 25 | 87 | 55.775 | 12.263 | 200 |
| Years of education | 0 | 16 | 7.885 | 2.423 | 200 |
| Family members (excluding the respondent) | 0 | 5 | 1.355 | 0.885 | 200 |
| Diversification | 0 | 1 | 0.660 | 0.475 | 200 |
| Diversification income | 0 | 500,000 | 20,028 | 55,362.39 | 200 |
| Inadequacy score | 0 | 0.733 | 0.150 | 0.158 | 177 |

With regard to the transfer of the land, again, only 191 individuals had the right to contract out their land to others (for them to farm on a temporary basis). Of these, 78 respondents contracted out their land, while 113 did not. This also corresponds to the fact that more than half of the respondents did not own houses in towns. Respondents who do not own houses in town may be seen as the permanent rural population, who are more inclined to manage both farmland and GfG forest land.

Women respondents made up 48% and men 52% out of the total 200 respondents, while 183 are married. Most respondents are currently married, and divorces in rural areas are rare. The mean of the number of children is 1.705. About 40% of respondents had two children, whereas 37.5% had only one child. The average age of respondents in the study areas was about fifty-six and the number of years of education was around eight years, meaning that on average, respondents did not finish junior high school. Excluding respondents, the number of household members consisted of 1.36 people. There were 2.35 people living in one household in rural areas on average, similar to the results of the Seventh National Census that found the average size of Chinese households to be 2.62 people [57].

More than half of the respondents have diversified incomes. However, the incomes of respondents who have diversified is 20,028 Yuan, including people who became better-off through diversification and people who had to diversify to survive. On the one hand, some respondents diversified their income by working for others, such as by washing dishes and harvesting. On the other hand, better-off respondents, such as restaurants owners or other entrepreneurs, diversified to enhance their entrepreneurial capability rather than for survival. Thus, the standard deviation shows a large number in our descriptive statistics. As diversification may explain the disparities between the incomes of the rural poor and the better off [58], we need to use it as an important control variable in statistical analysis. For the outcome variable, the mean of the inadequacy score was 0.150, which means that on average, respondents have inadequate achievements in 15% of domains (see Table 1).

### 3.1.2. Inadequacy Scores of Farmers

To complement the inadequacy scores in Table 3, we examined the results for all respondents who have scores in the five domains described in Section 2.2.2. Although the variable is continuous, respondents could have the same scores due to the unified calculation method. Table 4 indicates 14 inadequacy scores by gender. Around a third of respondents achieved full adequacy (inadequacy score = 0), with both men and women showing roughly the same proportion (32.26% and 32.14% for men and women, respectively). The male to female ratio is similar when there is no inadequacy on any indicator in the five domains (Table 1). Conversely, no respondent received a score equal to 1, which would reflect inadequacy for all 10 indicators. Nearly a quarter of respondents (about 18%) concentrated around a score of 0.267. Among them, 16.67% of women and 19.35% of men

have inadequate achievements in 26.7% of five domains. When the inadequacy score is greater than 0.3, however, the ratio of women is higher than men. In other words, there were more women than men who had inadequate achievements greater than 30% in the five domains.

**Table 4.** Inadequacy score by gender.

| | Male | Female | Total | | Male | Female | Total |
|---|---|---|---|---|---|---|---|
| | Freq. (%) | Freq. (%) | Freq. (%) | | Freq. (%) | Freq. (%) | Freq. (%) |
| 0 | 30 (32.26) | 27 (32.14) | 57 (32.20) | 0.3 | 2 (2.15) | 0 (0) | 2 (1.13) |
| 0.067 | 9 (9.68) | 8 (9.52) | 17 (9.60) | 0.333 | 0 (0) | 1 (1.19) | 1 (0.56) |
| 0.1 | 15 (16.13) | 6 (7.14) | 21 (11.86) | 0.367 | 1 (1.08) | 5 (5.59) | 6 (3.39) |
| 0.133 | 0 (0) | 1 (1.19) | 1 (0.56) | 0.433 | 0 (0) | 2 (2.38) | 2 (1.13) |
| 0.167 | 10 (10.75) | 8 (9.52) | 18 (10.17) | 0.533 | 2 (2.15) | 2 (2.38) | 4 (2.26) |
| 0.2 | 5 (5.38) | 6 (7.14) | 11 (6.21) | 0.633 | 1 (1.08) | 2 (2.38) | 3 (1.69) |
| 0.267 | 18 (19.35) | 14 (16.67) | 32 (18.08) | 0.733 | 0 (0) | 2 (2.38) | 2 (1.12) |
| Total | 177 (100) | | Male | 93 (100) | Female | | 84 (100) |

## 3.2. Factors Associated with Status of Men and Women

In Table 5, we displayed the FRM after margin results to check the correlation of the status of rural men and women with all possible influencing factors. The dependent variable was the inadequacy score for the five domains. Regarding participation in the GfG, the inadequacy score decreased 9.2% for a rural person who changed from being non-GfG to GfG (significant at the 1% level). GfG was negatively correlated with the inadequacy score, which may be due to the fact that families met a lot of problems in the process of conducting GfG, so they had to defend for themselves. Within the household, women would contribute their ideas to husbands and discuss them with each other. Both the husband and the wife solve problems, and decision-making powers are thought to be fairly evenly split.

**Table 5.** The correlation of the status of rural men and women with influencing factors.

| | dy/ex | Delta-Method SE |
|---|---|---|
| GfG (join GfG program) | −0.092 *** | 0.019 |
| Gender (male) | −0.045 ** | 0.013 |
| Marriage | −0.092 *** | 0.027 |
| Years of education | −0.014 *** | 0.005 |
| Age | 0.004 *** | 0.001 |
| Diversification (have diversified income sources) | 0.022 | 0.020 |
| Diversification income | $-9.54 \times 10^{-7}$ ** | $4.73 \times 10^{-7}$ |
| Number of children | 0.024 ** | 0.012 |
| Family size except respondents | 0.026 ** | 0.016 |
| House in town (own house in town) | −0.085 *** | 0.026 |
| Contracted land (transferred land) | −0.002 | 0.018 |
| Number of observations | 177 | |

***, ** denote 1% and 5% significance level, respectively.

To be specific, coefficients for gender show that males were associated with a 4.5% decrease in the inadequacy score. From this correlation, we can infer that rural women's

status overall is lower than rural men. This also confirms the results in Table 4, which show that when the score is greater than 0.3, there are more women. However, marriage was associated with a 9.2% decrease in the inadequacy score at a 1% significance level. Currently, marriage lowers the inadequacy score of rural residents across the five domains.

For the common control variables, we found that a 1% increase in age, number of children, and family size (excluding respondents) were associated with a 0.4% increase in the inadequacy score (significant at 1% level), a 2.4% increase in the inadequacy score (significant at 5% level), and a 2.6% increase in the inadequacy score (significant at 5% level), respectively. In Chinese households with couples who are more than 65 years of age, husbands tend to hold the purse strings and manage farm work by themselves. A 1% increase in years of education was associated with a 1.4% decrease in the inadequacy score (significant at 1% level). If rural people own a house in town, their inadequacy score decreased 8.5% at a 1% significance level. The correlation between the inadequacy score and contracted land was not that substantial, possibly because of regional differences.

From the perspective of diversification, the coefficient of having diversified income sources was not significant. It has little or almost no correlation with the inadequacy score, suggesting that diversification meets rural people's basic necessities, but not enough to change people's status. Although the coefficient of diversification income is significant at the 5% level to inadequacy score, the influence on people's status is minimal.

### 3.3. Average Treatment Effect of GfG on Rural Women's Status
#### 3.3.1. The Impacts of GfG on All Farmers

The findings presented in Table 6 revealed the average treatment effect of GfG on the inadequacy score and both treated and untreated potential outcome means (PO mean) for men and women, collectively and separately. From 177 respondents, the PO mean shows that the average inadequacy score for those who have not been treated is 0.180 at a 1% significance level, compared to the treated group with score 0.095 at a 1% significance level. A score of $-0.085$ (significant at 1% level) meant that participation in the GfG causes the inadequacy score to be reduced by an average of 0.085 points from the average of 0.180 points for farmers who did not participate in the GfG. Here, 0 is equal to adequate achievements, and 1 is equal to inadequate. The GfG was shown to have a significant effect on reducing inadequate achievements for each indicator in Table 1. It appears that the group with GfG is much better than the group without GfG in terms of acquiring adequate achievements across a set of different domains. Considering the first question of research, the results suggest that the status of men and women have generally risen.

**Table 6.** ATE results in inadequacy score by men and women, collectively and separately.

|  | Men & Women | | Men | | Women | |
|---|---|---|---|---|---|---|
|  | **Coef.** | **Robust SE** | **Coef.** | **Robust SE** | **Coef.** | **Robust SE** |
| ATE | $-0.085$ *** | 0.018 | $-0.035$ * | 0.021 | $-0.118$ *** | 0.027 |
| POMean |  |  |  |  |  |  |
| Without GfG | 0.180 *** | 0.014 | 0.150 *** | 0.016 | 0.215 *** | 0.022 |
| With GfG | 0.095 *** | 0.013 | 0.114 *** | 0.017 | 0.097 *** | 0.020 |
| Number of obs | 177 |  | 93 |  | 84 |  |

\*\*\*, \* denote 1% and 10% significance level, respectively.

Through the respondents' statements, we observed that participating in the GfG program cost the farmers money and time. Many different fees such as the design fee of afforestation operations are involved in engineering expenses and registration of forest rights certification, while there are long-time waiting times for government approval. One female respondent stated: "If I keep the status quo, our family cannot cut down trees (as we were not authorized). Instead, I want to make an active action on my property loss. I gathered others who have the same problems and reported that to project leaders".

Similarly, a male respondent said: "I have been searching the internet for information about how to apply for a certificate quickly as it is illegal to cut down trees without a certificate". Apart from the above, the program led to other unexpected developments. For example, a male respondent noted "The GfG trees blocked out the sunlight to the crops of adjacent farmers, leading to discontent among other farmers. My wife and I went to the village committee. We told committee members our plan to solve the problem. They mediated between other farmers". Meanwhile, a female farmer explained that "The GfG saplings were eaten by sheep, but we wanted to continue GfG, thinking of selling pine nuts for our elderly life. First, I asked for compensation from the owner of the sheep. The problem is the new saplings cannot be bought now. I am prepared to wait until next year when the seeds go on sale. I do not know what the outcome of the second try will be, but I still want to continue". Responses clearly show that farmers in the study area who participated in the GfG have more problems in terms of forestland management and ties with the outside community. Therefore, individuals must engage in more negotiations with relevant stakeholders. In this way, skills such as decision-making skills and problem-solving abilities at the individual level developed across the five domains, thus decreasing their inadequacy achievements.

### 3.3.2. The Impacts of GfG on Gender: The Differences of the Performance between Men and Women

As far as the second research question is concerned, which looks at men and women separately, the ATE on women is −0.118 (significant at 1% level) and on men is −0.035 (significant at 10% level). The results indicate that participation in GfG reduced inadequate achievements in each indicator by 11.8% and 3.5% for women and men, respectively. For women, participation in the GfG also had the largest declines in inadequacy in the five domains among the categories of "Men & Women", "Men" and "Women". On the other hand, if we compare the PO mean of respondents who did not participate in GfG, the women's inadequacy score (0.215 at a 1% significance level) was larger than that of men (0.150 at 1% significance level), indicating that women have more inadequate achievements in each indicator (or in other words, inferior decision-making power). A comparison of the PO mean of respondents who participated in GfG found that the women's inadequacy score (0.097 at a 1% significance level) was smaller than the men's score (0.114 at a 1% significance level), which means that women have more adequate achievements in each indicator. This suggests that women's decision-making power increased more than that of men after they participated in the GfG.

Taking advantage of the northeast region's topography, we found that some women in the study area sold wild plants picked in the mountains, while women who work in forest farms, for example, grow herbs under the forest and sell these medicinal materials by themselves. Two female restaurant owners had the same situation: "The restaurant is mainly run by me. My husband assists me". By contrast, male respondents seldom mentioned family welfare and talked more about the GfG program itself. Because of the gender division of labor, rural women are thus more inclined to engage in economic activities other than farming, such as off-farm entrepreneurship. In response to the program, women are actively acquiring knowledge on new livelihood strategies where they are exposed to new experiences. At the same time, women strengthen their own abilities, which makes their inadequacy scores decline more than men.

### 3.4. Propensity Score Matching with Calipers for Robustness Check

As an auxiliary analysis, a robustness check permits the researcher to objectively assert causal inference and whether the coefficients had changed too much using different methods [59]. A propensity score matching with calipers is good for the robustness of research, as not using calipers can lead to poor balance between treated and untreated subjects [60]. However, the literature does not adequately specify what is an appropriate caliper width, making it difficult to select the width. For instance, Refs. [61,62] failed to

explain why the caliper width 0.05 was used. Regarding the choice of the caliper width, Ref. [63] recommended that for observational studies, the optimal caliper width can be 0.2 of the standard deviation of the logit of the propensity score. Meanwhile, Ref. [60] suggested a caliper of 0.25 standard deviations (0.25SD) for a good balance, as the imbalance in the potential confounder of the effect of treatment between exposed and unexposed subjects is markedly reduced. To achieve a good balance with propensity score matching, we take the suggestion of 0.25SD (Equation (5)).

$$caliper = \sqrt{\frac{standard\ deviation\ of\ the\ logit\ of\ the\ propensity\ score}{4}} \tag{5}$$

Through the calculation, we know that the value of the caliper width is 0.08. Table 7 shows how the common support is operated by controlling the optimal caliper. Individuals in the treated group without an appropriate match are named as "off-support" while the others who find a suitable match are called "on-support" [64], and vice versa for the untreated group. The overlap in Figure 4 shows the common support area with two separate kernel density estimates. In our case, nine observations were dropped, while a further analysis of the regression coefficients of the interaction term was conducted for one hundred and sixty-eight observations (given in Section Robustness Check). Afterwards, we validated the robustness of our empirical results by having a higher degree of freedom (df) (Table 8). Moreover, some of the value of standard errors (SE) becomes smaller in Table 8 compared to Table 6. The SE for PO mean on women without GfG is 0.022 in Table 6 and 0.021 in Table 8, while the SE for PO mean on men with GfG is 0.017 in Table 6, but 0.013 in Table 8. As the value of SE decreases, the reliability of the results increases. More importantly, the effects are almost the same. Table 8 shows that the ATE of GfG on the inadequacy score of women is a negative number (0.115 − 0.234 = −0.119), while for men, it is also negative (0.095 − 0.166 = −0.071), which signifies that the inadequacy scores of both men and women are being reduced. At the same time, the results in Table 6 (−0.035 for men and −0.118 for women) and Table 8 (−0.071 for men and −0.119 for women) also show that the inadequacy score was reduced more for women than men for each indicator.

**Table 7.** Caliper common support.

| psmatch2: Treatment Assignment | psmatch2: Common Support Caliper (0.08384653) | | |
|---|---|---|---|
| | **Off Support** | **On Support** | **Total** |
| Untreated | 0 | 93 | 93 |
| Treated | 9 | 75 | 84 |
| Total | 9 | 168 | 177 |

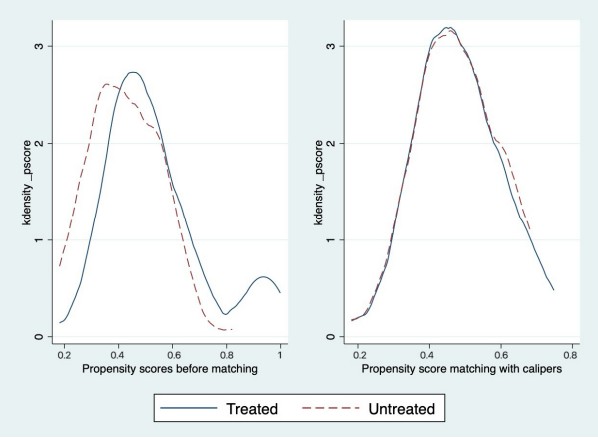

**Figure 4.** Caliper common support. Source: Authors.

**Table 8.** Marginal effects for interaction terms.

| | Margin | Delta-Method Std. Err. | $p > |z|$ |
|---|---|---|---|
| GfG#gender | | | |
| 0 0 | 0.234 | 0.021 | 0.000 |
| 0 1 | 0.166 | 0.017 | 0.000 |
| 1 0 | 0.115 | 0.020 | 0.000 |
| 1 1 | 0.095 | 0.013 | 0.000 |

## 4. Discussion

Drawing on the experiences from the GfG program, our study points to areas which are relevant for designing gender-responsive schemes of environment management in the future. Our study results accord with past studies that show that environmental policies have a positive relationship with women's empowerment [15,16]. Furthermore, we extended the concept of using WEAI to determine the status of men and women and found that women's status increased more than men. In this section, we discuss how environmental policies affected gender empowerment, as reflected in the differences in the changes in farmers' status by gender.

First, it is worth mentioning the eco-compensation included in environmental policies. Women have higher risk aversion than men in environmental policies [33]. Since the eco-compensation is often not enough to cover family expenses, women struggle to create new lives. Ref. [65] emphasized that women do this out of concerns about their family's welfare. This behavior change can increase women's empowerment through agency and achievements [66]. Second, in contrast to [35], who noted that the increase in natural resources can lead to an increase in women's decision-making power, the authors found that there was no increase in natural resources for women. The functional transformation of natural resources brought challenges to men and women. Through the process of facing different kinds of challenges, individuals may feel they can make decisions by themselves or get the results they desire [67]. People formulated a plan which they thought was the optimal approach, and carried out the plans, prepared to bear the consequences. Additionally, women and men in one household would take advice from each other and solve problems together. In this process, farmers have improved their ability to communicate and execute actions. Third, we partly agree with [18,34] that after GfG, women were under more pressure. The authors believed that the pressure was not just on women, but on both men and women in households. However, men and women focus on different aspects of environmental policies. Men focus on solving the problems that arise during the program, while women not only pay attention to the program, but also to the things outside the program, such as choosing to become an entrepreneur to diversify income sources. This is in line with the perspective of [59] that agricultural entrepreneurship should be underlined in agricultural diversification, as it creates more innovation, markets and resources. Environmental policies can bring opportunities for women's empowerment [34]; for example, having an independent income source may raise the social status of women and improve their negotiating position in household decisions and society.

Nevertheless, we also found other factors which may be associated with the change of rural people's status. Having more children, being older, and having more people living together may decrease rural people's status. Having more family members means women have to devote more time and energy to caregiving. Household stress like childcare and old care can have a psychological burden on women [68–70]. Older men and women are at high risk of future functional decline [71] as health and economic conditions continue to decline.

People who are currently married, have a high education level, and own a house in town may have a higher status in the five domains. After marriage, the wife has more opportunities to influence her husband's decisions regarding their shared property [72]. The

assets of spouses are tied together, and married men and women have increased property rights [73]. Besides marriage, education is very important as well. Many studies confirm that access to education can help improve social status. School education is a necessary tool for occupational success [74], and it is a cornerstone of an adult's socioeconomic status [75]. Last, owning a house in town reflects both economic status and the desire to improve the children's education. As mentioned earlier, rural people buy a house in the town so that the next generation can receive a good education in that town, which means home buyers pay attention to the education of the next generation. Therefore, it is not surprising that the results show them to have lower inadequacy scores, or in other words, a higher status.

## 5. Conclusions

This study evaluated the status of rural men and women collectively and separately from the perspective of participation in the GfG program in northeast China. At present, only a few studies in China have looked at the gender perspective of environmental policy, and most are qualitative in nature. Unlike previous studies, we combined our quantitative analysis with feedback from farmers to understand how the GfG program changed the status of men and women.

Through the empirical analysis of primary data, we made some new findings. Participation in the GfG program increased the status of rural men and women in northeast China, but women's status improved more than men's. The amount of time and effort spent on the project as well as unexpected results from the operational process made it difficult for farmers to manage forests well. They had to plan how to solve the problems. In the process of negotiating with the government sector and other stakeholders to defend their natural property rights, farmers' decisions in different domains were strengthened through information collection, communication, and plan execution. Importantly, the increase in women's status was notable, as it drew attention to the problem of livelihoods concern. Due to the impact on family welfare caused by the reduction of arable land, rural women were more likely than men to devote themselves to seeking extra income to make up for the loss arising from the GfG. Some rural women made great efforts to develop agricultural entrepreneurship, which ultimately led to strong decisions in their families and local society. This increase in status had not been anticipated in the GfG program; it was more the result of the need to diversify, which arose from the implementation of the GfG. It is hoped that the findings of this research will provide insights into gender differences for policymakers when determining the future direction of environmental preservation.

The current study has some limitations. First, this study was limited to the GfG Program implemented in the mountainous areas of the northeast region, and the results may therefore not apply to other regions. Second, it considered only the five domains of empowerment, without including other domains such as health and violence, which are perceived as important in studies such as The Status of Women in the States 2015 [76]. Third, our results are established on the assumption of conditional independence, or the selection on observables assumption.

Further research is needed to overcome the limitations mentioned above to better understand the issue. Continuing to investigate changes of women's statuses relative to men in the agricultural sector could strengthen environmental policies. When piloting environment-related programs, multiple effects such as economic impacts and different laws for both men and women should be monitored. The government could cooperate with academic institutions to ensure that both men and women have access to innovation, markets, and resources. For example, instead of limiting the program to providing eco-compensation, the government can support new jobs and training.

**Author Contributions:** Conceptualization, Y.Z. and K.L.M.; Methodology, Y.Z. and K.L.M.; Software, Y.Z.; Formal analysis, investigation, resources, data curation, Y.Z.; Validation, Y.Z. and K.L.M.; Writing—original draft preparation, Y.Z.; Writing—Revision, review, editing, Y.Z. and K.L.M.; Visualization, Y.Z. and K.L.M.; Supervision, K.L.M. All authors have read and agreed to the published version of the manuscript.

**Funding:** The author received some financial support to publish this article from the Hiroshima University Taoyaka Program for creating a flexible, enduring, peaceful society, funded by the Program for Leading Graduate Schools, Ministry of Education, Culture Sports, Science and Technology of Japan.

**Institutional Review Board Statement:** The study and questionnaire were approved by the Ethics Committee of Graduate School for International Development and Cooperation (IDEC), Hiroshima University, Japan on 2 March 2021. During and after the survey, a. we guaranteed respect for human rights; b. we obtained the approval from respondents; c. we ensured safety in all respects; and d. we protected personal information.

**Informed Consent Statement:** Informed consent was obtained from all respondents involved in the study.

**Data Availability Statement:** Data are unavailable in principle due to privacy. However, we accept the checks of raw data and Stata code with the consent of the authors.

**Conflicts of Interest:** The authors declare no conflict of interest.

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
