# Peer review of "Does Participation in the “Grain for Green Program” Change the Status of Rural Men and Women? An Empirical Study of Northeast China"

_sustainability, doi:10.3390/su152316447_

Round 1

Reviewer 1 Report

Comments and Suggestions for Authors

Those ideas could take this strong article to the next level. But overall, it is well-written and researched, and makes a valuable contribution. I would recommend only minor revisions before publication. Let me know if any of those suggestions need clarification!

Expand a bit more on the literature review and theoretical background. The introduction covers the empirical background on the Grain for Green program well, but is lighter on establishing the theoretical motivations regarding gender empowerment and environmental policies. Adding some more discussion of relevant theory could strengthen the framing.

Provide more details on the sampling and data collection process. The sampling method is described briefly but additional details on the survey administration, quality checks, response rates, etc. could add transparency.

Expand the conclusion/recommendations section. This is quite short currently. The authors could provide more concrete recommendations on integrating gender perspectives into environmental policies. Restating the key theoretical and empirical contributions would strengthen the concluding remarks.

Check for opportunities to streamline wording or reduce repetition between the background and discussion. Some parts cover similar ground and could be condensed.

Add subtitles within the results section for organizational clarity. This is a long section that could be broken up with descriptive subtitles.

Ensure the tables/figures are adequately discussed in the text and the results are interpreted. Some more narration connecting the empirical results to the research questions would help guide the reader.

Author Response

We attached our reply to this box.

Please kindly check it.

Thank you very much!

Yifei and K.L.Maharjan

Reviewer 2 Report

Comments and Suggestions for Authors

The paper has low quailiy.

Author Response

Dear Referee,

Could you offer more points?

Thank you for your cooperation!

Yifei and K.L.Maharjan

Reviewer 3 Report

Comments and Suggestions for Authors

Review comments:

The authors need to bring more clarity in the title – what status of men or women (social, decision making, power, or others?)

The abstract does not clarify what is the central inquiry/argument of the paper; it should include what the study contributes /suggests out of its findings.

Avoid using acronyms in the ‘key words’ section.

I think the Introduction section should start from line 59 with changes in description and correct and enhanced form of English. Then, literature review on payment for ecosystem services should be done on mountain ecosystems (or the category of ecosystems the study investigated), its biophysical characteristics in order to contextualize the problem/issue of the research in relation to the studied ecosystems (of Northeast China). In that case, line 1-58 can either go under 2.1 with a revised/different heading such as ‘Context and Study Sites’ or a new sub-section entitled ‘Study Context’ can be created by authors as 1.1.

 There are issues with in-text citations; for example, in line 67 citation [12], the author’s name should be specified; similar issue was found in line 147, and I think could be in some other places (so, please check the citation style/guide again).

In line 78, the authors need to use reference to justify their statement that “women’s benefits have been neglected”  

I think the paragraph comprising line 80-94 attempts to make the ‘Problem Statement’ for the research BUT it is largely confounded and poorly written. The authors must establish a clear ‘problematization’ of the article to proceed with its central argument and establish/state the objective of the research precisely.

I think lines 95 to 105 should go to the Methods section under Study sites

In line 107 specify what are the “other statistical model” – and justify with reference if such model was used by other studies and why.

Again, in the research questions (which serve as the research objectives as well) the authors need to clarify what they mean by ‘status’ OR else, they need to define the status before it is introduced in the article. As well, the authors must discuss with reference to contemporary literature how such status was measured elsewhere in the world, and what learnings and lessons are offered by such empirical examinations (measurements) of status.

As regards methods

-         How 112 HHs or .097% become representative of such a large population? The authors need to justify with statistical validity.

-        The study did not introduce why those 5 domains of empowerment were necessary to measure.

-        Why covid-19 impact was necessary to measure?

-        A questionnaire with 60 questions (involving semi-structured ones) is much larger than a standard set, and survey with it must be difficult to be accomplished in 30-60 minutes.

-        Why did the authors have to consider both treatment and control variables?

The Discussion section:

-        did not adequately flow from the study results. The results from statistical analysis should be discussed and referred to here in relation to the study objectives (research questions).   

-        Many qualitative quotes are included in the Discussion section; I am just lost, if the study adopted a mixed method, it should be specified beforehand in the methods section and the quotes must be included in the result part as well.

There is no need for a separate Limitation section. The authors can specify the study limitation within 2/3 sentences at the end of the Conclusions and then suggest what kind of further research should be carried out (that could be free from such limitations)

Overall, I find the research covered a lot of grounds BUY it lacks focus and a clear angle. In its present form, it is largely confounded and convoluted. 

Comments on the Quality of English Language

Professional English language editing would enhance the readability of the article.

Author Response

We attached our reply to this box.

Please kindly check it.

Also, the article was checked by a native English speaker who is the English editing specialist.

Thank you very much!

Yifei and K.L.Maharjan

Reviewer 4 Report

Comments and Suggestions for Authors

The Grain for Green program is an afforestation project created by the China Government aiming to protect the ecological environment. This manuscript is interesting, and very useful to understand the social effects of this program. However, there are some information needed to make it clear.

First, there are only 200 household heads of a total of 112 rural households in at least three cities. is it enough in the entire region?

Second, which city do they live in? could you add their distribution location in the manuscript.

In line 218, does it need to use figure 3 in the manuscript?

In the discussion section, please readjust this section to make it more concise and clear.

Comments on the Quality of English Language

In line 464-467, please check is it right of the cited references style? 

Author Response

(The authors gave the same response as above.)

Round 2

Reviewer 2 Report

Comments and Suggestions for Authors

The authors should improve the quality of the paper carefully.

Author Response

Thank you for your comments.

We improved the contents especially on the theoretical motivations regarding women and environment to strengthen the article logic.

Yifei and K.L.Maharjan

Reviewer 3 Report

Comments and Suggestions for Authors

I appreciate that the comments are adequately addressed. Is not it better to term the last section as 'Conclusion' rather than 'Summary and Conclusion'?

Comments on the Quality of English Language

I think the English editing services is required to make the overall articulation more tightened and fluid.

Author Response

Thank you for your comments.

We changed the title of last section to "conclusion".

Also, we shortened limitation to make it more streamlined.

Yifei and K.L.Maharjan

Reviewer 4 Report

Comments and Suggestions for Authors

This form is fine.

Author Response

Thank you for your comments.

Yifei and K.L.Maharjan